# Simple Development of Novel Reversible Colorimetric Thermometer Using Urea Organogel Embedded with Thermochromic Hydrazone Chromophore

**Tawfik A. Khattab** [1], **Mehrez E. El-Naggar** [2,*], **Meram S. Abdelrahman** [1], **Ali Aldalbahi** [3] **and Mohammad Rafe Hatshan** [3]

1 Dyeing, Printing and Auxiliaries Department, Textile Industries Research Division, National Research Dokki, P.O. Box 12622, Giza 12522, Egypt; tkhattab@kent.edu (T.A.K.); abdelrahman.meram@yahoo.com (M.S.A.)
2 Pre-Treatment and Finishing of Cellulose Based Textiles Department, Textile Industries Research Division, Dokki, P.O. Box 12622, Giza 12522, Egypt
3 Department of Chemistry, College of Science, King Saud University, Riyadh 11451, Saudi Arabia; aaldalbahi@ksu.edu.sa (A.A.); mhatshan@ksu.edu.sa (M.R.H.)
* Correspondence: mehrez_chem@yahoo.com

**Abstract:** Thermochromic urea (U) organogel immobilized with a thermochromic tricyanofuran hydrazone (TCFH) chromophore was developed. Thermochromic TCFH chromophore bearing two nitro functional groups on a hydrazone recognition unit was synthesized via an azo-coupling reaction of tricyanofuran (TCF) heterocyclic moiety containing an active methyl group with the diazonium chloride salt of 2,4-dinitroaniline comprising two strongly electron-withdrawing nitro groups. The molecular structure of both intermediates and TCFH dye were characterized by several analytical methods, including $^1$H NMR, $^{13}$C NMR, IR, mass spectroscopy (MS), and elemental analysis. The thermochromic responsiveness could be attributed to the charge delocalization of TCFH as well as to the presence of an intramolecular charge transfer. The generated organogel displayed a thermoreversible sol–gel transition associated with color change. The origin of the monitored thermochromism is a conformational change of the tricyanofuran hydrazone backbone due to the temperature-driven deprotonation–protonation reversible process. The prepared urea–tricyanofuran hydrazone (UTCFH) thermometer acted as a diagnostic tool providing an instant color change between yellow, orange, red and purple upon changing the temperature of the UTCFH organogel in dimethyl sulfoxide (DMSO). This color change was proportionally correlated with increasing the temperature from 44 to 63 °C. The UTCFH organogel composed of urea and push-$\pi$-pull hydrazone type tricyanofuran chromophore immobilized physically in the urea organogel was found to function as a temperature-driven chromic thermometer. This chromogenic UTCFH organogel in DMSO displayed a phase transition at 41–48 °C. The morphological properties of the gel internal fibrous nanostructure (80–120 nm) were monitored by scanning electron microscopy (SEM). The colorimetric measurements were monitored by UV–Vis absorption spectroscopy. The chromogenic thermometer demonstrated a good reversibility without fatigue. The mechanism accounting for thermochromism of UTCFH organogel is proposed.

**Keywords:** urea; tricyanofuran hydrazone; smart organogel; nanofibers; thermochromic; colorimetric

## 1. Introduction

The development of colorimetric sensors as environmentally monitoring early warning tools is highly interesting due to their ability for simple and fast naked-eye detection without the necessity for trained personnel, or any expensive or sophisticated instruments [1–3] Therefore, colorimetric sensors have recently attracted more interest in the monitoring of various hazardous materials [4–14]. The simple synthesis of arylhydrazone dyes is an effective process for their versatile chromogenic properties. Hydrazone dyes are of high significance due to their physico-chemical properties and potential applications, such as sensors, nonlinear optics, and medical purposes. This can be ascribed to their unique molecular structure, which is highly influenced by the type and location of substituents [15,16]. Tricyanofuran-based dyes with push-π-pull molecular systems have drawn more interest due to their nonlinear optical characteristics. They have demonstrated a high sensitivity to environmental changes, such as pH of the medium and solvent polarity [11,17–20]. Tricyanofuran hydrazones have been synthesized as excellent optical materials in different applications, such as fluorescent and colorimetric sensors, electro-optics, single molecule fluorophore, photo-refractive, logic memories, dye-based lasers and organic light-emitting devices [21,22]. Tricyanofuran hydrazone dyes have significant characteristics, including photo- and thermal stability and comparatively high yield, as well as their capability to provide a wide range of colors depending on the substituents located on the arylhydrazone unit [23]. They have been applied for use as antimicrobial agents and disperse dyestuffs for polyester fibers as well as solvatochromic, biochromic, metallochromic, vapochromic, thermochromic and halochromic sensors. Their solvatochromic activity is valuable as potential polarity sensors for molecular electronics, solvents, in the fabrication of solutions that absorb light at a certain frequency, and to investigate the conformations and binding of proteins. Tricyanofuran hydrazone dyes have been applied as environmental chemosensors for bacterial strains as well as for pH and temperature changes [24]. The performance of tricyanofuran hydrazones is attributed to their push-π-pull intramolecular charge transfer character. Upon deprotonation, the hydrazone recognition moiety functions as an electron π-bridge connecting electron-push and electron-pull units or it may function itself as an electron-push unit when existing in a conjugated structure with strong electron-withdrawing substituent(s). The phenylhydrazone unit substituted with strong electron-withdrawing substituent(s) has an acidic secondary amine (NH) group with the ability to produce a hydrazone nitrogen anion, after proton abstraction, associated with improved electron donation ability. Conjugation of the generated hydrazone anion with highly electron-pull moiety should result in potentially interesting vapochromic, metallochromic, solvatochromic, thermochromic, solvatofluorochromic, halochromic and biochromic dyes [21–24].

Temperature is the most commonly employed physical stimulus in environmentally thermoresponsive materials [25–27]. The development of a colorimetric-based thermochromic sensor device or chromogenic thermometer has been a desired property to monitor temperature changes [28–30]. Thermochromism has been employed for many practical applications, such as baby bottles, kettles changing color at the boiling point of water, aquariums, inks, refrigerators, medical purposes, and indicators to determine the temperature changes for different industrial purposes. One of the distinctive characteristics of thermochromic materials is the existence of a critical solution temperature. Organogels have been used as thermoreversible viscoelastic materials composed of a large quantity of solvent molecules and a nanofibrous entangled network [31–34] generated by the self-assembly of organogelator molecules [35,36]. Organogels have a variety of potential applications, such as sensors, biomedical devices, energy harvesting and optoelectronics [37–40]. The distinct ability of organogels to immobilize solvents extends interest towards tunable physical properties and responsiveness to external stimuli. Stimuli-responsive organogels have been recognized as interesting smart materials due to their significant applications, such as removal of pollutants, drug delivery, as well as tissue engineering [41–45]. Therefore, the development of stimuli-responsive organogels has been very appealing. George et al. recently reported the preparation of structurally simple low molecular-weight organogelators from urea derivatives with one or two *n*-alkyl substituents at the

nitrogen atoms. As a result of strong intermolecular hydrogen bonding, the produced organogels were thermally reversible and necessitate very low concentrations (<2 wt%) of the urea gelator [46].

In this context, we report on the fabrication of a simple-structured thermochromic organogel-based colorimetric thermometer using urea as a thermoresponsive organogel and a tricyanofuran hydrazone as a thermochromic chromophore. The tricyanofuran-based arylhydrazone chromophore containing two strongly electron-withdrawing nitro groups was easily synthesized via azo coupling of tricyanofuran heterocycle containing an active methyl group with the diazonium salt of 2,4-dinitroaniline. The synthesized tricyanofuran hydrazone dye was physically embedded into urea gel. The thermal deprotonation, polarizability, and responsiveness of the tricyanofuran hydrazone molecular system were improved by the insertion of strongly electron-withdrawing nitro groups onto the phenylhydrazone moiety. This colorimetric thermometer is characterized by its ability for simple and fast naked-eye monitoring, as well as high sensitivity without the necessity for any complicated electric equipment or electronic parts. The morphology of the generated organogel was studied by scanning electron microscopy (SEM). The thermochromic properties of the urea–tricyanofuran hydrazone (TCFH) organogel were explored by UV–Vis absorption spectra. The thermochromic mechanism of urea–tricyanofuran hydrazone (UTCFH) was proposed based on structural differences of TCFH. The presented results demonstrate a new mechanism to visualize temperature changes by the naked eye, providing a novel advance towards the development of thermochromic materials. Furthermore, the existence of alkaline vapors results in a remarkable shift in the chromogenic performance, indicating possible applications in fields such as gaseous sensors as well as drug delivery.

## 2. Experimental Details

### 2.1. Materials and Reagents

Chemicals and reagents employed in this study, including dicyanomethane, 3-hydroxy-3-methylbutan-2-one, 2,4-dinitraniline and urea, were purchased from commercial sources (Sigma-Aldrich) and were applied without any additional purification. Solvents (spectroscopic grade), urea (98%) and *n*-octadecylamine were obtained from Aldrich. *n*-Octadecylamine (distilled twice under vacuum) was employed in the synthesis of *N*-*n*-octadecylurea according to previously reported literature procedure [47]. Tricyanofuran intermediate and hydrazone probe were prepared according to a previous reported method [46,47]. The reaction profiles were monitored by Merck aluminum thin layer chromatography (TLC) sheets pre-coated with silica-gel PF-254, which were visualized under UV lamp at 254/365 nm. To guarantee high purity, the synthesized TCFH chromophore and tricyanofuran (TCF) intermediate were subjected to flash column chromatography and re-crystallization, respectively. Experimental results were collected under ambient conditions if not stated otherwise.

### 2.2. Apparatus and Methods

Melting points and thermal stability were measured by differential scanning calorimetry (DSC; TA-2920) and thermogravimetric analysis (TGA; TA-2950), respectively. Elemental analysis (C, H, N) was investigated by PerkinElmer2400 (Norwalk, United States). Fourier-transform infrared (FT-IR) spectra were reported by Bruker-Vectra-33 with attenuated total reflection (ATR) probe. Mass spectra were reported by Shimadzu GC-MS-QP 1000EX at 70 eV. Nuclear magnetic resonance ([1]H and [13]C NMR) spectroscopy was measured by BRUKER/AVANCE at 400 MHz under ambient conditions; chemical shifting was presented in ppm units relative to the tetramethylsilane internal standard. The temperature-dependent [1]H NMR spectra were studied by gradually heating a solution of tricyanofuran hydrazone (0.50 mM) in DMSO-$d_6$. UV–Vis absorption spectra were recorded on an HP8453 spectrophotometer. The morphology was studied by scanning electron microscopy (SEM) (Hitachi S2600N) at 20 kV. The diameter of the generated xerogel fibers was measured using software J image associated with SEM. The organogel fabricated from urea immobilized with tricyanofuran hydrazone probe in DMSO was poured onto a glass substrate and left to dry under vacuum in a

desiccator. After that, the generated xerogel was annealed at 45 °C for 12 h. The surface area was determined using Quantachrome TouchWin software version 1.21 and NOVAtouch surface area analyzer (USA) under nitrogen gas adsorption–desorption isotherms. The sample was degassed in vacuum under ambient conditions over 3 h. The Brunauer–Emmett–Teller (BET) technique was employed to determine the specific surface area.

## *2.3. Synthetic Methods*

### 2.3.1. Synthesis of Tricyanofuran (TCF)

In a 500 mL round bottom flask positioned in a water-bath, Na metal (600 mg, 16 mmol) was added into absolute ethanol (20 mL) under ambient conditions. Both dicyanomethane (24 g, 362 mmol) and 3-hydroxy-3-methylbutan-2-one (18.0 gm, 176 mmol) were then added into the as-prepared $CH_3ONa$ solution with stirring for 1 h. An additional amount of absolute ethanol (60 mL) was added, followed by reflux for 1 h. The mixture was subjected to cooling in a fridge; the generated precipitate was filtered off, washed with cold ethanol (20 mL), and finally air-dried to provide off-white powder (17.05 g). The filtrate was concentrated under vacuum to afford extra off-white powder (2.01 g); the total yield was 57%; mp 201–203 °C. The purity of the synthesized tricyanofuran was confirmed by TLC and NMR spectra; $^1$H NMR (400 MHz, $CDCl_3$): ppm 2.36 (s, 3 H), 1.65 (s, 6 H).

### 2.3.2. Synthesis of *N-n*-Octadecylurea

In a 100 mL round bottom flask, a mixture of urea (120 mg, 2 mmole) and *n*-octadecylamine (270 mg, 1 mmole) was magnetically stirred at 160 °C for 3 h. After cooling, the admixture was crushed and heated with distilled water to remove the unreacted urea. The product was then crystallized twice from chloroform to provide 155 mg; m.p. 105–108 °C; theoretical elemental analysis for $C_{19}H_{40}N_2O$ (312.31): C 73.02, H 12.90, N 8.96; found: C 73.41, H 13.15, N 8.57.

### 2.3.3. Synthesis of Tricyanofuran Hydrazone Probe (TCFH)

In a 100 mL Erlenmeyer flask, a mixture of 2,4-dinitroaniline (10 mmol), HCl (5 mL) and distilled water (5 mL) was stirred in an ice-bath at 0–5 °C, followed by slowly adding sodium nitrite (12 mmol) dissolved in distilled water. The generated diazonium salt was stirred for an extra 15 min at 0–5 °C. The produced diazonium chloride cold solution was then added slowly to a mixture of acetic acid (4 mL), sodium acetate (3 g) and the as-prepared tricyanofuran intermediate (10 mmol) was dissolved in acetonitrile at 0–5 °C in a separate 100 mL Erlenmeyer flask. The crude product was filtered off under vacuum; rinsed with distilled water (40 mL) and subjected to re-crystallization from *n*-propanol/chloroform (1:1 ratio) to give light red powder (yield 59%). The TCFH purity was proved by TLC and NMR spectra. m.p. monitored at 235–237 °C; $^1$H NMR (400 MHz, DMSO-$d_6$): 12.13 (s-broad, 1 H, N-H), 8.58 (s, 1 H, =C-H), 8.38 (d, 1 H, *J* = 2.4 Hz), 8.23 (dd, 1 H, *J* = 11.6, 2.4 Hz), 7.65 (d, 1 H, *J* = 8.8 Hz), 1.82 (s, 6 H); $^{13}$C NMR (400 MHz, DMSO-$d_6$): 176.81, 171.13, 142.03, 132.25, 125.95, 124.54, 115.45, 112.57, 111.81, 110.69, 25.80; IR (neat, $\nu/cm^{-1}$) 3256 (N-H), 2226 (CN), 1577 (C=N), 1509 and 1324 ($NO_2$); MS *m/z* (%): 392 $[M - H]^+$; theoretical elemental analysis for $C_{17}H_{11}N_7O_5$ (~393): C 51.91, H 2.82, N 24.93; found: C 52.78, H 2.65, N 24.77.

## *2.4. Preparation and Gelation Procedure of UTCFH*

To study the effect of TCFH concentration on the gelation properties, the concentration of TCFH in the UTCFH–DMSO mixture was changed in the range between 0.01, 0.02, 0.04, 0.06, 0.08, 0.10, 0.15 and 0.20 wt%, while the urea concentration was fixed at 2 wt%. The organogel formation course was carried out by the dissolution of urea immobilized with tricyanofuran hydrazone probe in DMSO. The mixture was then heated up from room temperature (25 °C) to 65 °C at a rate of 2 °C/min. Under ambient conditions, the admixture was cooled back to room temperature to introduce the organogel. The organogel was generated in a time-period of ~8–13 min, which was related to the

TCFH concentration. The melting temperature of the organogel was measured according to the test tube inversion technique, in which the vial enclosing organogel was positioned upside down in a paraffin oil bath. This oil bath was heated up at a rate of 2 °C/min. Upon heating, the temperature at which the gel falls down under gravity was reported as the organogel melting point.

### 2.5. Thermochromic Measurements

The thermochromic behavior of UTCFH in DMSO solution at pH = ~6.65 was monitored by increasing the temperature from 25 to 65 °C at a heating rate of 2 °C/min, while recording the UV–Vis absorption spectra.

### 2.6. Reversibility Testing

The reversibility of UTCFH in DMSO solution at pH value of ~6.65 was examined by changing the temperature back and forth between 25 and 65 °C by increasing the temperature to 65 °C at a heating rate of 2 °C/min; and then cooling back under ambient conditions to 25 °C over several cycles while recording the maximum absorption wavelength for each cycle at 44 and 63 °C. The pH values of the different solutions were monitored by Beckman Coulter pHI340 with a glass calomel electrode. The images of the vials containing tricyanofuran hydrazone probe at different pH values were taken using Canon Power Shot A-710-IS digital camera.

## 3. Results and Discussion

### 3.1. Synthesis and Characterization

The straightforward synthesis procedure for the dinitro-substituted tricyanofuran hydrazone is depicted in Scheme 1. The tricyanofuran intermediate was synthesized according to previous literature approach [48,49]. It should be useful to illustrate the correlated sides of the current anticipated chemistry for the synthesis of tricyanofuran–hydrazone (TCFH) probe and to elucidate its relationship with the present objective of producing a novel reversible colorimetric thermochromic organogel. Knoevenagel condensation has been essential in the synthesis of electron-deficient moieties, such as the current strongly electron-withdrawing tricyanofuran (TCF) moiety that has been applied in various donor–acceptor molecular systems. In a wide range of chemical reactions, the active methyl, and methylene-containing compounds such as dicyanomethane or $\alpha$-nitriles bearing carbonyl functional groups can interact with a base such as sodium ethoxide in order to undergo an azo-coupling with diazonium chloride salts of aromatic amines or to undergo a condensation reaction with aldehydes or ketones. The one-pot sequent reaction with an additional equivalent of the active methylene and/or methyl-containing adducts followed by ring-closing results in strongly electron deficient heterocyclic moieties as shown in Scheme 2.

A hydrogen atom located next to electron deficient substituents in a certain molecular system is highly acidic. The electron-withdrawing substituents assist with stabilizing the carbanion generated via proton abstraction from an activate methyl or methylene group by a base. The synthesis was simply carried out via an azo-couple reaction of tricyanofuran heterocycle with aryl diazonium salt in the presence of a base to give the analogous unstable azo-containing intermediate, which shifts straight to the corresponding hydrazone-containing molecular structure characterized by higher stability. The hydrazone NH in the tricyanofuran–hydrazone spectroscopic probe was proved by $^1$H NMR displaying a signal at 12.13 ppm, and by IR represented through a band at 3256 cm$^{-1}$. The hydrazone N=CH group displayed a broad signal at 8.58 ppm. The IR spectrum showed bands at 2226 and 1577 cm$^{-1}$ assigned for the cyano (CN) and C=N groups, respectively. Two characteristic absorption bands were monitored at 1509 and 1324 cm$^{-1}$ due to the nitro (NO$_2$) groups [48,49].

**Scheme 1.** Synthesis of allyloxy-substituted tricyanofuran hydrazone spectroscopic probe.

**Scheme 2.** Two-step one-pot Knoevenagel condensation to generate tricyanofuran heterocycle.

### 3.2. Thermal Stability of Tricyanofuran Hydrazone Dye

The thermal behavior of the dinitro-substituted tricyanofuran hydrazone chromophore was gauged by differential scanning calorimetry (DSC) and thermal gravimetric analysis (TGA). The dinitro-substituted tricyanofuran hydrazone chromophore contains two strongly electron-withdrawing nitro substituents on the phenylhydrazone group showing a thermal stability up to ~236 °C (decomposition temperature).

### 3.3. Preparation of UTCFH Thermometer

The tricyanofuran hydrazone probe exhibited an excellent solubility in DMSO [34]. On the other hand, the low molecular-weight organogelator from a simple and inexpensive *n*-alkyl-substituted urea at the nitrogen atoms displayed good organogelation properties in DMSO mainly due to strong intermolecular hydrogen bonding. Furthermore, the urea gelator bearing an alkyl $n$-$C_{18}H_{37}$ (Figure 1) chain demonstrated good organogelation properties compared to other urea organogelator derivatives [31]. Hence, the urea gelator was selected as a host to immobilized tricyanofuran hydrazone probe as a guest in DMSO as a solvent. The organogel was tested by dissoluting urea immobilized with tricyanofuran hydrazone probe in DMSO by heating at 65 °C to afford a clear solution (sol) associated with a color change to purple. This was followed by cooling the mixture back to room temperature to introduce the organogel (gel) associated with its original yellow color. This temperature-responsive sol–gel cycle was totally reversible without fatigue. The organogel was generated in a time-period of 10–25 min depending on the gelator (urea) concentration. The organogel melting temperature was measured according to the test tube inversion technique.

The effect of tricyanofuran hydrazone probe concentration on the gelation performance was explored. Thus, TCFH was applied at a range of different concentrations including 0.01, 0.02, 0.04, 0.06, 0.08, 0.10, 0.15 and 0.20 wt%, while urea concentration was fixed at 2 wt%. The thermal stability of UTCFH gel in DMSO was studied by recording the TCFH concentration dependent gel→sol transition temperature at a certain pH value (~6.65) as displayed in Figure 2. It was monitored that the gel melting point was slightly decreased between 48 and 41 °C upon increasing the concentration of TCFH

between 0.01 and 0.08 wt%, respectively. Nonetheless, no gelation performance was monitored at a TCFH concentration equal to or higher than 0.08 wt%. In other words, the organogelation process was disturbed by increasing the number of TCFH molecules between urea molecules. The rationale for the decreased organogel thermal stability at a total TCFH content value higher than 0.08 wt% could be attributed to decreasing the stability of the generated intermolecular hydrogen bonding between urea molecules by the immobilized TCFH molecules. The effect of humidity on the gelation properties was also explored. The generated gels demonstrated a high sensitivity to water, and consequently to humidity. The addition of a trace amount (0.01 wt%) of distilled water was highly sufficient to liquefy the organogel. Similarly, the sample organogel will often melt upon exposure to atmospheric humidity for a few hours. This behavior strongly proves that H-bonding is the driving force for the self-assembly and formation of the organogel. The action of water, and consequently humidity, in disrupting the organogel can then be ascribed to the disruption of the H-bonds by water molecules [50,51].

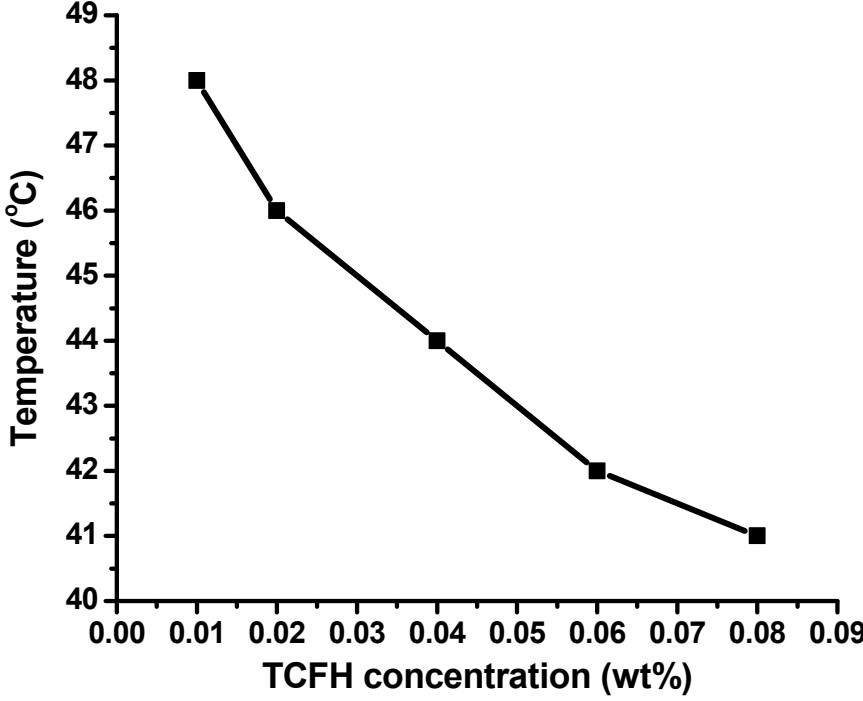

**Figure 1.** Molecular structure of urea organogelator.

**Figure 2.** Melting point of urea–tricyanofuran hydrazone (UTCFH) in DMSO (pH = ~6.65) upon increasing tricyanofuran hydrazone (TCFH) concentration.

### 3.4. Thermochromic Activity

The UV–Vis absorption spectra of UTCFH were found to change in the range of 442–535 nm upon increasing the temperature (Figure 3). This can be ascribed to the highly electron-withdrawing nitro groups at both *ortho* and *para* positions of the phenylhydrazone [48,49]. Hence, this phenylhydrazone is able to generate an acidic NH to release a proton resulting in the formation of a dinitro-substituted

phenylhydrazone anion with an improved electron donating ability of the dinitro-substituted phenylhydrazone. The strongly electron-withdrawing nitro groups themselves cannot function as electron-donors. However, the phenylhydrazone moiety is capable of functioning as an electron π-bridge in a donor–acceptor system or functioning itself as an electron-donor once conjugated with a strongly electron-withdrawing group, such as the nitro substituent (Scheme 3). Thus, this nitro-substituted phenylhydrazone anion acts as an electron-donor conjugating with TCF moiety as an electron-acceptor leading to potentially interesting spectral switch properties. The thermochromic sol–gel processing of UTCFH (0.06 wt%) in DMSO is demonstrated in Figure 4.

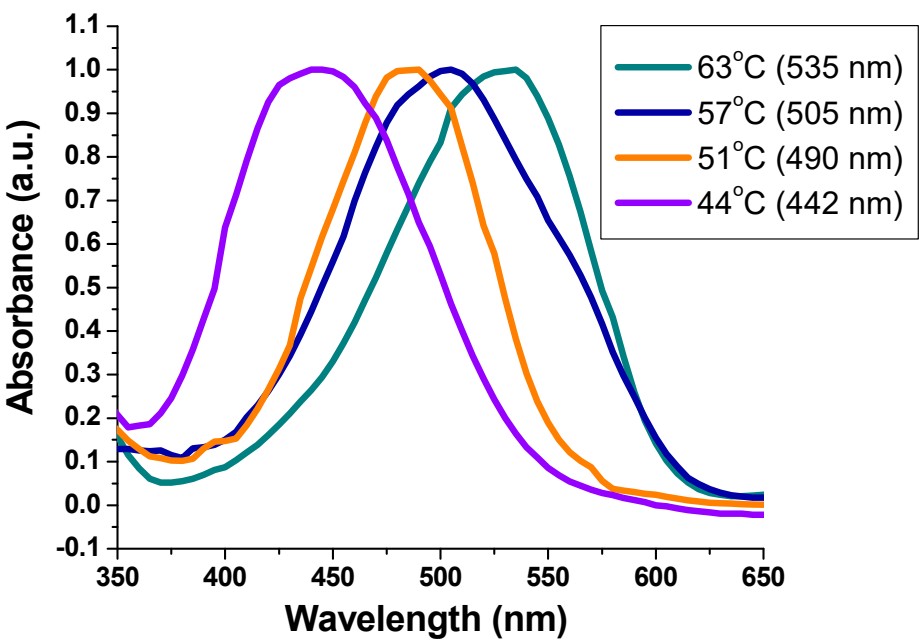

**Figure 3.** Normalized UV–Vis absorbance spectra of UTCFH (0.06 wt%) in DMSO at different temperature values; 44 °C (yellow at 442 nm), 51 °C (orange at 490 nm), 57 °C (red at 505 nm), and 63 °C (purple at 535 nm).

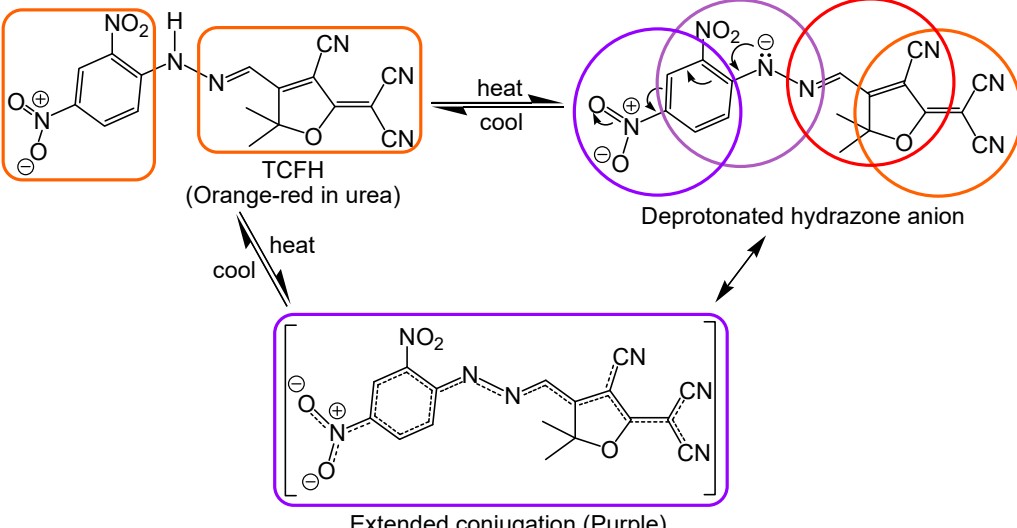

**Scheme 3.** Temperature-driven molecular switching mechanism of tricyanofuran hydrazone spectroscopic probe embedded in urea organogel.

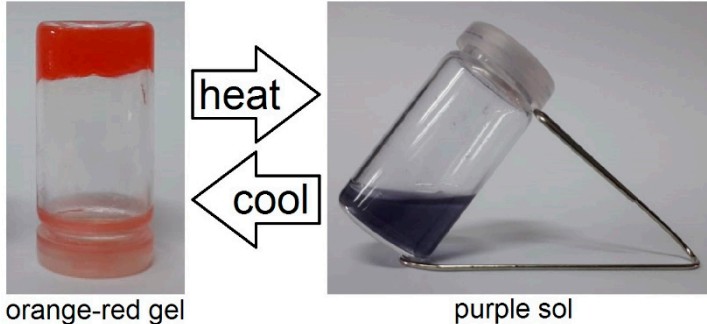

**Figure 4.** Illustration of thermochromic sol–gel process of UTCFH (0.06 wt%) in DMSO.

The purple color of UTCFH was found to reversibly change to yellow upon cooling in dimethyl sulfoxide (DMSO) solution. No similar color shift was monitored with the urea organogel alone under the same condition but in the absence of the tricyanofuran hydrazone probe. Obviously, the generated hydrazone anion is essential for the thermochromism of UTCFH. The induced planarity of the nitro-substituted tricyanofuran hydrazone anion is higher than this with the tricyanofuran hydrazone chromophore itself. Thus, the thermochromism occurs via an equilibrium sequence among two molecular isomers undergoing a temperature-driven deprotonation–protonation reversible process. The temperature-driven deprotonation of TCFH results in an extended conjugation of the nitro-substituted tricyanofuran hydrazone anion, which is obviously longer than that of the neutral tricyanofuran hydrazone itself. This is favorable to the electron transfer leading to the color shift from yellow to purple.

To inspect the reversibility of this UTCFH against temperature changes, the temperature of UTCFH in DMSO solution at pH = ~6.65 was changed back and forth by increasing the temperature from 25 to 65 °C while recording the maximum absorption wavelength at 44 and 63 °C. The solution was then left to cool back to room temperature at 25 °C to regenerate the organogel. This procedure was repeated twelve times while recording the absorption spectra for each cycle. The stability of the maximum absorption wavelengths at 442 and 535 nm was recorded, and the results are depicted in Figure 5. No significant changes in the maximum absorption wavelengths were monitored at 442 and 535 nm. Thus, it is apparent that UTCFH exhibited high stability and high reversibility without fatigue.

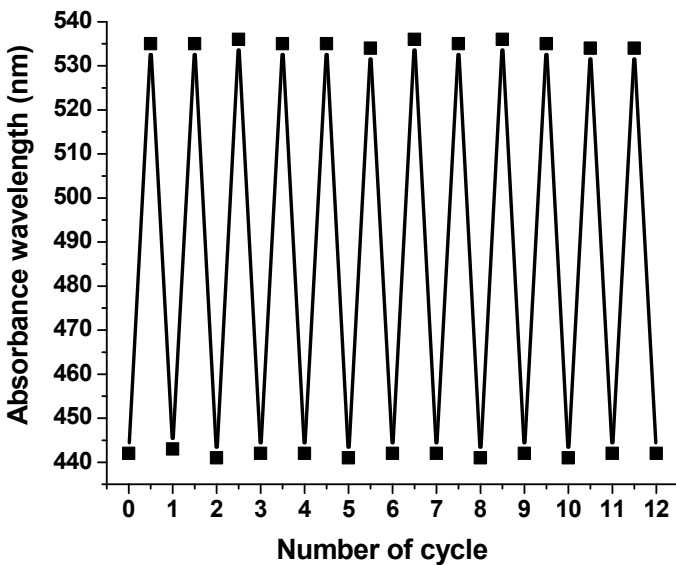

**Figure 5.** Changes between the absorption maximum wavelengths of UTCFH (0.06 wt%) at 442 and 535 nm at the temperature values of 44 and 63 °C, respectively; pH value was adjusted to ~6.65.

The thermochromism of organic materials generally occurs via an equilibrium sequence among two isomers. The extended conjugation of the deprotonated formyl-substituted hydrazone anion $\pi$-system is obviously longer than that of the hydrazone form. Thus, the color changes from yellow to purple. The deprotonated nitro-substituted hydrazone anion form was proved by both of IR and $^1$H NMR spectra. The $^1$H NMR spectra were recorded at both room temperature (25 °C) and 65 °C to confirm that the secondary NH hydrazone peak (at 12.13 ppm) disappeared upon increasing temperature to 65 °C (Figures S1–S5, S7 and S8, Supplementary Materials). Additionally, the signals at 8.58 (=C-H), 8.38 (aromatic CH), 8.23 (aromatic CH), 7.65 (aromatic CH) and 1.82 (aliphatic CH) demonstrated slight shifts to 8.50, 8.31, 8.20, 7.66 and 1.84, respectively. The distinctive secondary NH hydrazone peak (at 3256 cm$^{-1}$) was found to disappear in the IR spectrum of the deprotonated hydrazone anion form (Figures S6 and S9). To study the IR spectrum of the hydrazone anion form, the solid nitro-substituted hydrazone chromophore was dissolved in a mixture of triethylamine/acetone (1:1) and then air-dried under ambient conditions. The peaks at 2226 (CN), 1577 (C=N), 1509 and 1324 (NO$_2$) showed some slight shifts to 2210, 1587, 1509 and 1328, respectively.

### 3.5. Morphological Properties

The self-assembly process of UTCFH led to the generation of nanofibrous supramolecular architectures. The morphology of the UTCFH xerogel (after complete solvent evaporation) was explored with a scanning electron microscope (SEM). From the SEM images, the three dimensional entangled nanofibrous network with a fiber diameter between 80–120 nm is shown in Figure 6a,b. This supramolecular architecture of the xerogel obviously illustrates the three-dimensional network generated by intertwined nanofibers that become swollen when immobilized by DMSO to form the organogel. The tricyanofuran hydrazone chromophore was physically embedded in the organogel matrix. The continuous addition of the tricyanofuran hydrazone chromophore (0.01–0.08 w%) within the organogel matrix influenced the xerogel porosity demonstrating that the chromophore was completely immobilized in the nanofibrous film matrix (Figure 6a,b). Thus, no organogel was assembled upon increasing the concentration of the chromophore from 0.08 to 0.2 wt% (Figure 6c,d). This could be ascribed to disrupting the H-bonding by the increased total content of TCFH molecules, which in turn disrupt the self-assembly process. This was also proved by evaluating the specific surface area of the produced organogels employing the Brunauer–Emmett–Teller (BET) approach. The nitrogen gas adsorption–desorption isotherm was applied using Quantachrome TouchWin version 1.21. H-bonding is the driving force for the self-assembly and formation of the organogel. The surface area of the prepared organogels was found to decrease in the range of 248, 203, 175, 136, 84, 69, 51 and 33 m$^2$ g$^{-1}$ with increasing TCFH concentration in the range of 0.01, 0.02, 0.04, 0.06, 0.08, 0.10, 0.15 and 0.20 wt%, respectively. This could be attributed to disrupting the H-bonds by the increased total content of TCFH molecules [50,51]. The created large surface area could offer more active sites, and consequently a higher surface sensing sensitivity. These results are in concurrence with the morphological results obtained by SEM. Interestingly, the xerogel was found to be reversibly responsive to alkaline vapors with an instant color shift from yellow to orange, red and purple. Thus, those nano-scale fibers may demonstrate high sensitivity in sensing alkaline analytes due to their high surface area and highly porous architecture, which facilitates the convenient diffusion of analytes through the fibrous xerogel matrix. As expected, the xerogel film can be applied as a portable quick *onsite* detection technique to monitor toxic industrial gases, such as toxic colorless ammonia in both gaseous and liquid phases. The produced UTCFH xerogel obtained from DMSO with a total content of TCFH at 0.06 wt% was air-dried and drop-casted (deposited) onto a filter paper strip to display a color change from orange to purple upon exposure to gaseous ammonia (Figure 7). The fabricated urea-based xerogel film demonstrated a fibrous morphology. The three-dimensional fibrillated nanoporous scaffold was assembled from the urea derivative gelator embedded with the tricyanofuran hydrazone thermochromic probe. It demonstrated a large surface area, highly porous architecture, and high sensitivity to gaseous molecular ammonia due to its simplistic diffusion through the mesh of the

xerogel matrix to reach active TCFH sites. The current chromogenic thermometer is characterized by a simple preparation procedure using inexpensive materials, including the biodegradable urea and a very low concentration of TCFH chromophore. An instant change between different colors was monitored from yellow to orange, red and purple at a relatively wide temperature range from 44 to 63 °C. Additionally, it demonstrated a good reversibility without fatigue. However, the current process still requires improvement to increase the detectable temperature range to between 35–100 °C, which will be useful in monitoring human temperature and other physiological processes at 37 °C, as well as monitoring various industrial applications at high temperatures.

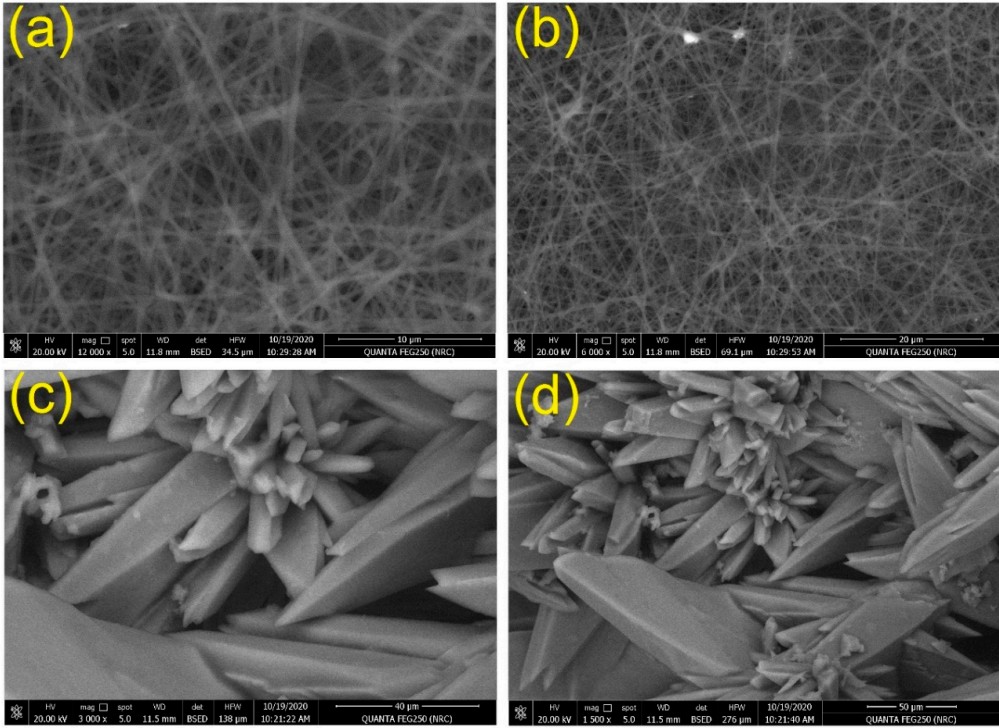

**Figure 6.** SEM images of the dried UTCFH xerogels obtained from DMSO with a total content of TCFH at 0.06 wt% (**a**,**b**), and 0.2 wt% (**c**,**d**).

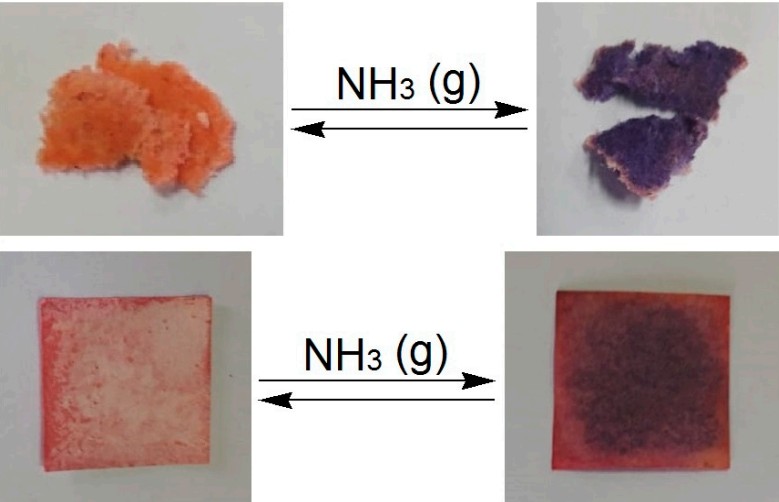

**Figure 7.** Air-dried UTCFH xerogel (**top**) as well as UTCFH xerogel deposited onto a filter paper strip (**bottom**) obtained from DMSO with a total content of TCFH at 0.06 wt% demonstrating a color change from orange to purple upon exposure to gaseous ammonia.

## 4. Conclusions

We demonstrated a reversible thermochromic organogel consisting of nitro-substituted tricyanofuran hydrazone chromophore uniformly immobilized into urea in DMSO to generate a thermoresponsive organogel. The UV–Vis absorption spectra of the organogel markedly changed between yellow, orange, red and purple during the thermoreversible sol–gel transition. This was attributed to a temperature-responsive tricyanofuran-based spectroscopic probe bearing a hydrazone sensing moiety as part of a π-conjugate molecular system. An attractive temperature-driven deprotonation–protonation process of tricyanofuran hydrazone immobilized in urea organogel was monitored. The urea organogel was not able to change color alone; however, UTCFH was able to self-assemble to generate a thermochromic organogel with high thermal stability and its morphology showed nanofiber architectures with a diameter of about 80–120 nm.

**Supplementary Materials:** The following are available online at http://www.mdpi.com/2227-9040/8/4/132/s1, Figure S1: [1]H NMR spectra of TCF heterocyclic molecule. Figure S2: [1]H NMR spectra of TCFH chromophore (hydrazone form) under ambient conditions. Figure S3: [1]H NMR spectra of TCFH chromophore (hydrazone form) under ambient conditions; magnification of aliphatic area. Figure S4: [1]H NMR spectra of TCFH chromophore (hydrazone form) under ambient conditions; magnification of aromatic area. Figure S5: [1]H NMR spectra of TCFH chromophore (hydrazone form) under ambient conditions demonstrating the signal for the NH group. Figure S6: FT-IR spectra of TCFH chromophore (hydrazone form). Figure S7: [1]H NMR spectra of TCFH chromophore (hydrazone anion form) at 65°C. Figure S8: [1]H NMR spectra of TCFH chromophore (hydrazone anion form) at 65 °C; magnification of aromatic area. Figure S9: FT-IR spectra of TCFH chromophore (hydrazone anion form); the solid powder of the hydrazone form was dissolved in a mixture of triethylamine/acetone (1:1) and then air-dried under ambient conditions to generate the hydrazone anion form.

**Author Contributions:** Conceptualization and investigation, T.A.K., M.E.E.-N., A.A.; Methodology, T.A.K., M.E.E.-N., A.A., M.S.A., M.R.H.; writing—original draft preparation, T.A.K., M.E.E.-N., A.A.; supervision, T.A.K., M.E.E.-N., A.A. All authors have read and agreed to the published version of the manuscript.

**Funding:** This research was funded by Researchers Supporting Project number (RSP-2020/30).

**Acknowledgments:** The authors acknowledge King Saud University, Riyadh, Saudi Arabia, for funding this work through Researchers Supporting Project number (RSP-2020/30).

**Conflicts of Interest:** We wish to confirm that there are no known conflicts of interest associated with this publication. We confirm that the manuscript has been read and approved by all named authors and that there are no other persons who satisfied the criteria for authorship but are not listed. We further confirm that the order of authors listed in the manuscript has been approved by all of us.

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
