# Peer review of "Simple Development of Novel Reversible Colorimetric Thermometer Using Urea Organogel Embedded with Thermochromic Hydrazone Chromophore"

_chemosensors, doi:10.3390/chemosensors8040132_

Round 1

Reviewer 1 Report

The authors reported unique method to progress a reversible thermochromic organogel with assistance well-known component tricyanofuran hydrazone. The manuscript is well-written and the result is worthy of publication. However, some statements still require further justification.

Please, clarify the speed of temperature changing during the experiments of investigation any thermo- characteristics.

It may be interesting to add humidity among the working parameters of novel organogels.

It was mentioned " It demonstrated a large surface area, highly porous architecture and high sensitivity to gaseous molecular ammonia due to its simplistic diffusion through the mesh of the xerogel matrix to reach TCFH active sites" but in results and discussion section you did not give the full answer and proofs, if your proposition gives the solution of those problems. What do you mean ‘high sensitivity”?

How about the reversibility and responsiveness for the present material, both may be affected by adsorbed surface square less than thickness/diameter of fibers?

Author Response

Point-by-point response to Reviewers’ comments

Date: October 25th, 2020

Manuscript ID: chemosensors-964161

Type of manuscript: ArticleTitle: Facile development of novel reversible colorimetric thermometer using urea organogel embedded with thermochromic hydrazone chromophore

Chemosensors

Dear Professor

Editor

Chemosensors

We are really thankful for giving us the opportunity to revise our manuscript entitled above. I carefully considered the reviewers’ comments. I want to extend my appreciation for taking the time and effort necessary to provide such insightful guidance. The revision, based on the review team’s collective input, includes a number of positive changes. Based on your guidance, I have accordingly modified the manuscript and detailed corrections, changes and/or rebuttals against each point raised are listed below. I hope that these revisions improve the paper such that you and the reviewers now deem it worthy for publication in Chemosensors.

With the submission of this manuscript, we would like to undertake that:

  • We have read and approved the final version submitted;
  • The contents of this manuscript have not been copyrighted or published previously;
  • The contents of this manuscript are not now under consideration for publication elsewhere;
  • The contents of this manuscript will not be copyrighted, submitted, or published elsewhere, while acceptance by Chemosensors is under consideration;
  • There are no directly related manuscripts, published or unpublished, by the author of this paper;
  • Our institutions’ representatives are fully aware of this submission.

The authors declare no conflict of interest.

Please address all correspondence concerning this manuscript to me at [email protected]

Thank you for your consideration!

Kind regards,

Mehrez El-Naggar, PhD

Textile Research Division

National Research Centre

Dokki, Cairo 12622, Egypt

Comments and Suggestions for Authors

The authors reported unique method to progress a reversible thermochromic organogel with assistance well-known component tricyanofuran hydrazone. The manuscript is well-written and the result is worthy of publication. However, some statements still require further justification.

Please, clarify the speed of temperature changing during the experiments of investigation any thermo- characteristics.

Authors’ Response: Thanks for the reviewer recommendation. The speed of temperature changing was clarified in manuscript. The organogel formation course was carried out by the dissolution of urea immobilized with tricyanofuran hydrazone probe in DMSO. The temperature of the mixture was then raised from room temperature (25 °C) to 65 °C at a heating rate of 10 °C/min. Under ambient conditions, the mixture was cooled back to room temperature to introduce the organogel.

It may be interesting to add humidity among the working parameters of novel organogels.

Authors’ Response: Thanks for the reviewer comment. The effect of humidity on the gelation properties was also explored was in manuscript. The generated gels demonstrated a high sensitivity to water, and consequently to humidity. The addition of trace amounts of distilled water was highly sufficient to liquefy the organogel. Similarly, the sample organogel will often melt upon exposure to atmospheric humidity for a few hours. This behavior strongly proves that H-bonding is the driving force for the self-assembly and formation of the organogel. The action of water, and consequently to humidity, in disrupting the organogel can then be ascribed to the disruption of the H-bonds by water molecules.

It was mentioned " It demonstrated a large surface area, highly porous architecture and high sensitivity to gaseous molecular ammonia due to its simplistic diffusion through the mesh of the xerogel matrix to reach TCFH active sites" but in results and discussion section you did not give the full answer and proofs, if your proposition gives the solution of those problems. What do you mean ‘high sensitivity”?

Authors’ Response: Thanks for the reviewer recommendation. The surface area was studied, and added to manuscript, using Quantachrome TouchWin software version 1.21 and NOVAtouch surface area analyzer (USA) under nitrogen gas adsorption-desorption isotherms. Brunauer-Emmet-Teller (BET) method was used to calculate the specific surface area. The sample was degassed in vacuum under ambient conditions during 3 hours. Nitrogen gas adsorption-desorption isotherm was applied using Quantachrome TouchWin version 1.21. The specific surface area was evaluated using Brunauer-Emmett-Teller (BET) approach. H-bonding is the driving force for the self-assembly and formation of the organogel. The surface area of the prepared organogels was found to decrease in the range of 248, 203, 175, 136, 84, 69, 51 and 33 m2 g-1 with increasing TCFH concentration in the range of 0.01, 0.02, 0.04, 0.06, 0.08, 0.10, 0.15 and 0.20 wt%, respectively. This could be attributed to disrupting the H-bonds by the increased total content of TCFH molecules. The created large surface area could offer more active sites, and consequently a higher surface sensing sensitivity. These results are in concurrence with the morphological results obtained by SEM.

How about the reversibility and responsiveness for the present material, both may be affected by adsorbed surface square less than thickness/diameter of fibers?

Authors’ Response: Thanks for the reviewer recommendation. Nitrogen gas adsorption-desorption isotherm was applied to explore the specific surface area using Brunauer-Emmett-Teller (BET) approach. H-bonding is the driving force for the self-assembly and formation of the organogel. The surface area of the prepared organogels was found to decrease in the range of 248, 203, 175, 136, 84, 69, 51 and 33 m2 g-1 with increasing TCFH concentration in the range of 0.01, 0.02, 0.04, 0.06, 0.08, 0.10, 0.15 and 0.20 wt%, respectively. This could be attributed to disrupting the H-bonds by the increased total content of TCFH molecules. The created large surface area could offer more active sites, and consequently a higher surface sensing sensitivity. These results are in concurrence with the morphological results obtained by SEM.

Reviewer 2 Report

The paper reported that thermochromic tricyanofuran hydrazone (TCFH) chromophore was immobilized within the urea organogel. The modified organogel system is used to detect the change of temperature sensitively in the range of 44 and 63 ̊C, and demonstrated good reversibility without fatigue. The results are good and very interesting. The paper should be published after the revised the following questions.

  • In Fig.(3), the UV-Vis absorption spectra of UTCFH are changed in the range of 442-530 nm with the increase of the temperature. If the UTCFH is using to the thermometer, the data are lacking. Please supplement.
  • Lines 248-249, the authors claim that no gelation performance was monitored at a TCFH concentration equal to or higher than 0.08 wt%. Why? Please give the evidence.
  • Please give the original spectra, such as NMR, IR, and add in the supporting information.

Author Response

Point-by-point response to Reviewers’ comments

Date: October 25th, 2020

Manuscript ID: chemosensors-964161

Type of manuscript: ArticleTitle: Facile development of novel reversible colorimetric thermometer using urea organogel embedded with thermochromic hydrazone chromophore

Chemosensors

Dear Professor

Editor

Chemosensors

We are really thankful for giving us the opportunity to revise our manuscript entitled above. I carefully considered the reviewers’ comments. I want to extend my appreciation for taking the time and effort necessary to provide such insightful guidance. The revision, based on the review team’s collective input, includes a number of positive changes. Based on your guidance, I have accordingly modified the manuscript and detailed corrections, changes and/or rebuttals against each point raised are listed below. I hope that these revisions improve the paper such that you and the reviewers now deem it worthy for publication in Chemosensors.

With the submission of this manuscript, we would like to undertake that:

  • We have read and approved the final version submitted;
  • The contents of this manuscript have not been copyrighted or published previously;
  • The contents of this manuscript are not now under consideration for publication elsewhere;
  • The contents of this manuscript will not be copyrighted, submitted, or published elsewhere, while acceptance by Chemosensors is under consideration;
  • There are no directly related manuscripts, published or unpublished, by the author of this paper;
  • Our institutions’ representatives are fully aware of this submission.

The authors declare no conflict of interest.

Please address all correspondence concerning this manuscript to me at [email protected]

Thank you for your consideration!

Kind regards,

Mehrez El-Naggar, PhD

Textile Research Division

National Research Centre

Dokki, Cairo 12622, Egypt

Comments and Suggestions for Authors

The paper reported that thermochromic tricyanofuran hydrazone (TCFH) chromophore was immobilized within the urea organogel. The modified organogel system is used to detect the change of temperature sensitively in the range of 44 and 63 ̊C, and demonstrated good reversibility without fatigue. The results are good and very interesting. The paper should be published after the revised the following questions.

In Fig.(3), the UV-Vis absorption spectra of UTCFH are changed in the range of 442-530 nm with the increase of the temperature. If the UTCFH is using to the thermometer, the data are lacking. Please supplement.

Authors’ Response: Thanks for the reviewer recommendation. The thermochromic (thermometer) behavior of UTCFH in DSMO solution at pH = 6.65 was monitored by increasing the temperature from 25 to 65°C at a heating rate of 2°C/minute, while recording the UV-Vis absorption spectra. These data are included in manuscript.

Lines 248-249, the authors claim that no gelation performance was monitored at a TCFH concentration equal to or higher than 0.08 wt%. Why? Please give the evidence.

Authors’ Response: Thank you so much for the reviewer recommendation. To study the effect of TCFH concentration on the gelation properties, the concentration of TCFH in the UTCFH-DMSO mixture was changed in the range between 0.01, 0.02, 0.04, 0.06, 0.08, 0.10, 0.15 and 0.20 wt%, while urea concentration was fixed at 2 wt%. The organogel formation course was carried out by the dissolution of urea immobilized with tricyanofuran hydrazone probe in DMSO. The mixture was then heated up from room temperature (25°C) to 65°C at a rate of 2°C/minute. Under ambient conditions, the mixture was cooled back to room temperature to introduce the organogel. The organogel was generated in a time period of ~8-13 minutes, which was related to the TCFH concentration. The melting temperature of the organogel was measured according to the test tube inversion technique, in which the vial enclosing organogel was positioned upside down in a paraffin oil-bath. This oil bath was heated up at a rate of 2°C/minute. Upon heating, the temperature at which the gel falls down under gravity was reported as the organogel melting point.

Please give the original spectra, such as NMR, IR, and add in the supporting information.

Authors’ Response: Original spectra were added to supporting information.

Thank you very much!

Reviewer 3 Report

The submission by Tawfik and coworkers reports on the development of a reversible colorimetric thermometer using a urea organogel embedded with a thermochromic hydrazone chromophore. After careful analysis of this work, this reviewer is of the opinion that the authors have not adequately provided evidence to substantiate some of the claims which are made and have not provided adequate evidence to prove the colorimetric switching mechanism for this material beyond simply citing previous literature. The burden is on the authors to prove experimentally that the mechanism of colorimetric switching is due to deprotonation such than their experimental results matches with previous literature. It is not scientifically sound to simply cite previous work to do this or to make claims of experimental observations for which such data is not provided in the submission. For example, the authors write on page 8, in response to the experimental observation of a UV-Vis absorption change in the range of 442-535 nm that, "Hence, this phenylhydrazone is able to generate an acidic NH to release a proton resulting in the formation of a dinitro-substituted phenylhydrazone". The evidence for the chemical transformation that the authors are claiming is not simply born out the observation of changes in the UV-Vis absorption spectra! In this case they simply cite a paper, yet provide no experimental evidence that the proton release has occurred. Furthermore, the authors write on page 11, "The deprotonated nitro-substituted hydrazone anion form was proved by both of IR and 1H NMR spectra. The distinctive secondary NH hydrazone peak (at 3256 cm-1) was found to disappear in the IR spectrum. The 1H NMR spectra was recorded at both room temperature (25 oC) and 65 oC to confirm that the secondary NH hydrazone peak (at 12.13 pm) diasppeared upon increasing temperature." This reviewer would like to point out to the editor and to the authors that THIS EXPERIMENTAL DATA IS NOT PROVIDED IN THE SUBMISSION! The supplementary information provided by the authors consists of only 2 NMR spectra (and 3 zoom in spectra) for 2 compounds and a single IR absorption spectra. All of this data is submitted without captioning and without proper labeling of the proton or IR signals to match with the chemical structures. Moreover, there is no experimental evidence that variable temperature NMR and variable temperature IR (at 25oC and 65oC as claimed) was actually carried out showing the disappearance of the signals at 12.13 ppm and 3256cm-1 . Are we to take the authors word that such experimental observations were actually made? The authors have even failed to included the nitrogen gas absorption isotherms, and their BET modeled fits for the data presented on page 12. Are these numbers fabricated? What are the mathematical equations used to model the gas absorption isotherm, if they in fact exist? This reviewer is not seeking to attack the credibility of the authors, but as scientists were are burdened with providing the experimental evidence in our peer reviewed manuscripts that justify our claims. In this case the lack of experimental evidence is directly tied to the very mechanistic claims of how this material operates, without which, the submission cannot stand on its own and should be rejected. 

There is yet even more questions unanswered. The authors claim in the original submission that , "the continuous addition of the trycyanofuran hydrazone chromophore (0.01-0.08%) within the organogel matrix did not influence the xerogel porosity....". However in the revised submission, the BET data found that the surface area of the material decreased from 248 to 84 m^2g^-1 when concentration of the chromophore was increased from 0.01 to 0.08%. So if increasing the concentration of the trycyanofuran doesnt influence the xerogel porosity, how is that BET analysis clearly shows substantial decreases in the surface area with increasing concentration of the chromophore? 

For a journal such as Chemosensors, the one experiment that actually involves chemo-sensing, the color changes due to ammonia adsorption, are poorly characterized. For such experiments, the chromic change over time is needed to gauge the kinetics of the color change and to ascertain whether the adsorption follows psuedo first order or pseudo second order kinetics. Isotherms of adsorption should also be conducted in order to evaluate whether we are looking a chemi-adsorption or physi-adsorption for the material and whether the process involves monolayer adsorption or multilayer adsorption. 

In view of all the issues outlined above, this author must recommend rejection of this manuscript as in its current form, there is a substantial lacking of experimental evidence to justify the claims, and the materials are inadequately characterized. This will inevitably prevent the community at large from testing the veracity of the claims and reproducing the experimental observations independently.  

Author Response

Point-by-point response to Reviewers’ comments

Point-by-point response to Reviewers’ comments

Comments and Suggestions for Authors

The submission by Tawfik and coworkers reports on the development of a reversible colorimetric thermometer using a urea organogel embedded with a thermochromic hydrazone chromophore. After careful analysis of this work, this reviewer is of the opinion that the authors have not adequately provided evidence to substantiate some of the claims which are made and have not provided adequate evidence to prove the colorimetric switching mechanism for this material beyond simply citing previous literature. The burden is on the authors to prove experimentally that the mechanism of colorimetric switching is due to deprotonation such than their experimental results matches with previous literature. It is not scientifically sound to simply cite previous work to do this or to make claims of experimental observations for which such data is not provided in the submission. For example, the authors write on page 8, in response to the experimental observation of a UV-Vis absorption change in the range of 442-535 nm that, "Hence, this phenylhydrazone is able to generate an acidic NH to release a proton resulting in the formation of a dinitro-substituted phenylhydrazone". The evidence for the chemical transformation that the authors are claiming is not simply born out the observation of changes in the UV-Vis absorption spectra! In this case they simply cite a paper, yet provide no experimental evidence that the proton release has occurred. Furthermore, the authors write on page 11, "The deprotonated nitro-substituted hydrazone anion form was proved by both of IR and 1H NMR spectra. The distinctive secondary NH hydrazone peak (at 3256 cm-1) was found to disappear in the IR spectrum. The 1H NMR spectra was recorded at both room temperature (25 oC) and 65 oC to confirm that the secondary NH hydrazone peak (at 12.13 pm) diasppeared upon increasing temperature." This reviewer would like to point out to the editor and to the authors that THIS EXPERIMENTAL DATA IS NOT PROVIDED IN THE SUBMISSION! The supplementary information provided by the authors consists of only 2 NMR spectra (and 3 zoom in spectra) for 2 compounds and a single IR absorption spectra. All of this data is submitted without captioning and without proper labeling of the proton or IR signals to match with the chemical structures. Moreover, there is no experimental evidence that variable temperature NMR and variable temperature IR (at 25oC and 65oC as claimed) was actually carried out showing the disappearance of the signals at 12.13 ppm and 3256cm-1 . Are we to take the authors word that such experimental observations were actually made?

Authors’ response: Thank you so much for the reviewer detailed feedback. Both NMR and FTIR spectra were added to the supplementary file demonstrating that the distinctive secondary NH hydrazone signal disappearing. Both Captioning and proper labeling of proton and IR signals were added to the images of NMR and FTIR spectra. However, the authors would like mention that the main target of this study is to develop a novel reversible colorimetric thermometer using urea organogel embedded with thermochromic hydrazone chromophore. Thus, the authors focused on exploring the novel production of the urea/hydrazone colorimetric thermometer. The hydrazone chromophore itself was previously studied in the literature and the authors did not want to repeat what was already studied in the previous literature. Also, the authors would like to get into the attention of the reviewer that we addressed all the previous concerns requested by the reviewer in the first revision and all the current comments are new and were not reported/requested by the reviewer in the first revision.

The authors have even failed to included the nitrogen gas absorption isotherms, and their BET modeled fits for the data presented on page 12. Are these numbers fabricated? What are the mathematical equations used to model the gas absorption isotherm, if they in fact exist? This reviewer is not seeking to attack the credibility of the authors, but as scientists were are burdened with providing the experimental evidence in our peer reviewed manuscripts that justify our claims. In this case the lack of experimental evidence is directly tied to the very mechanistic claims of how this material operates, without which, the submission cannot stand on its own and should be rejected. 

Authors’ response: As we mentioned in the manuscript, the surface area was determined using Quantachrome TouchWin software version 1.21 (new version) and NOVAtouch surface area analyzer (USA) under nitrogen gas adsorption-desorption isotherms. The sample was degassed in vacuum under ambient conditions during 3 hrs. Brunauer-Emmet-Teller (BET) technique was employed to determine the specific surface area. This is a new version of software and the value of surface area was automatically calculated by the instrument. Thus, there is no equation needed. I believe also that it will be unacceptable if we mentioned that we are using a certain equation. Bellow, there are some articles published in high rank journals studying the surface area under the same methodology without including an equation unless they added a fabricated data to their articles as the reviewer mentioned.

  • Krejčová, Ludmila, Terza Leonhardt, Filip Novotný, Vilém Bartůněk, Vlastimil Mazánek, David Sedmidubský, Zdeněk Sofer, and Martin Pumera. "A Metal‐Doped Fungi‐Based Biomaterial for Advanced Electrocatalysis." Chemistry–A European Journal 25, no. 15 (2019): 3828-3834.
  • Barad, Chen, Giora Kimmel, Hagay Hayun, Dror Shamir, Michael Shandalov, Gal Shekel, and Yaniv Gelbstein. "Influence of galia (Ga 2 O 3) addition on the phase transformations and crystal growth behavior of zirconia (ZrO 2)." Journal of Materials Science 53, no. 18 (2018): 12741-12749.
  • Hebert, Sabrina C., and Klaus Stöwe. "Synthesis and Characterization of Bismuth-Cerium Oxides for the Catalytic Oxidation of Diesel Soot." Materials 13, no. 6 (2020): 1369.
  • Mazánek, Vlastimil, Laura Pavlikova, Petr Marvan, Jan Plutnar, Martin Pumera, and Zdenek Sofer. "Fluorine saturation on thermally reduced graphene." Applied Materials Today 15 (2019): 343-349.
  • Rodrigues, Mônica A., Ariadne C. Catto, Elson Longo, Edson Nossol, and Renata C. Lima. "Characterization and electrochemical performance of CeO2 and Eu-doped CeO2 films as a manganese redox flow battery component." Journal of Rare Earths 36, no. 10 (2018): 1074-1083.

There is yet even more questions unanswered. The authors claim in the original submission that , "the continuous addition of the trycyanofuran hydrazone chromophore (0.01-0.08%) within the organogel matrix did not influence the xerogel porosity....". However in the revised submission, the BET data found that the surface area of the material decreased from 248 to 84 m^2g^-1 when concentration of the chromophore was increased from 0.01 to 0.08%. So if increasing the concentration of the trycyanofuran doesnt influence the xerogel porosity, how is that BET analysis clearly shows substantial decreases in the surface area with increasing concentration of the chromophore? 

Authors’ response: Thanks for the reviewer comments. The typographical error was fixed.

For a journal such as Chemosensors, the one experiment that actually involves chemo-sensing, the color changes due to ammonia adsorption, are poorly characterized. For such experiments, the chromic change over time is needed to gauge the kinetics of the color change and to ascertain whether the adsorption follows psuedo first order or pseudo second order kinetics. Isotherms of adsorption should also be conducted in order to evaluate whether we are looking a chemi-adsorption or physi-adsorption for the material and whether the process involves monolayer adsorption or multilayer adsorption. 

Authors’ response: Thanks for the reviewer recommendation. The main target of the current research manuscript, as demonstrated in both manuscript title and abstract, is the development of novel reversible colorimetric thermometer using urea organogel embedded with thermochromic hydrazone chromophore. Thus, the ammonia sensing process was not a major target for the current research manuscript as indicated in both title and abstract. However, it was mentioned in the manuscript as a result of our observations. Authors decided to study the chromic changes as well as psuedo first order or pseudo second order kinetics in a future work. Also, in the future work we will study the isotherms of the adsorption process to evaluate whether it is a chemi-adsorption or physi-adsorption; monolayer adsorption or multilayer adsorption. 

In view of all the issues outlined above, this author must recommend rejection of this manuscript as in its current form, there is a substantial lacking of experimental evidence to justify the claims, and the materials are inadequately characterized. This will inevitably prevent the community at large from testing the veracity of the claims and reproducing the experimental observations independently.  

Authors’ response: Thank you so much for the reviewer recommendation. We wish that the reviewer is satisfied and we addressed all his concerns. The authors just wanted to focus on the novel parts of the current research manuscript and to avoid repeating what was already studied in the previous literature. Also, the authors would like to get into the attention of the reviewer that we addressed all the previous concerns he requested in the first revision and all the current comments are new and were not reported/requested by the reviewer in the first revision.

Reviewer 4 Report

The authors investigated an excellent thermal sensor platform.  The manuscript described details of all the required parameters. However, to improve further the manuscript, I recommend the following corrections:

  1. Need to write clearly the need and applications of the thermosensors in the introduction section.
  2. Comparison and lacunas of current thermo sensors need to be discussed well.
  3. What is the effect of sensing due to humidity conditions?
  4. Figure 3 if possible gives data all the points of nm (350-650), then the fig will appear very smooth.
  5. If the author adds absorption maxima for each temperature in Fig. 3, it will increase the information from Fig. 3.
  6. The authors gave a very nice mechanism in scheme 3, if possible authors can give the graphical abstract as well.
  7. How many times/cycles the same solution shows reversibility for hot/cool conditions.
  8. Fig. 5 legend is not clear at all, what is meant by different temperature values, need to mention it very clearly.
  9. Have authors identified the mechanism with some structural validation, means structural changes in orange-red in urea, deprotonated hydrazone anion, and extended conjugation (purple) adducts
  10. This is a nice piece of work, well written, well presented, and highly important study.

Author Response

Manuscript ID: chemosensors-964161

Title: Facile development of novel reversible colorimetric thermometer using urea organogel embedded with thermochromic hydrazone chromophore

Authors: Tawfik Khattab, Mehrez E. El Nagar, Meram Abdelrahman, Ali Aldalbahi, Mohammad Rafe Hatshan

The authors investigated an excellent thermal sensor platform.  The manuscript described details of all the required parameters. However, to improve further the manuscript, I recommend the following corrections:

  1. Need to write clearly the need and applications of the thermosensors in the introduction section.

Response: the need and applications of thermosensors was described clearly in the introduction section (page 2; line 76-78). Thermochromism has been employed for many practical applications, such as baby bottles, kettles changing color at the boiling point of water, aquariums, inks, refrigerators, medical purposes, and indicators to determine the temperature changes for different industrial purposes.

  1. Comparison and lacunas of current thermo sensors need to be discussed well.

Response: Comparison and lacunas of current thermo-sensor were discussed in manuscript (see page 12, line 376-383). The current chromogenic thermometer is characterized by a simple preparation procedure using inexpensive materials, including the biodegradable urea and a very low concentration of TCFH chromophore. An instant change between different colors was monitored from yellow to orange, red and purple at a relatively wide temperature range from 44 to 63°C. Additionally, it demonstrated a good reversibility without fatigue. However, the current process still need for improvement to increase the detectable temperature range between 35-100°C, which will be useful to monitor human temperature and other physiological processes at 37°C, as well as monitoring various industrial applications at high temperatures.

  1. What is the effect of sensing due to humidity conditions?

Response: The effect of humidity on the gelation properties was also explored. The generated gels demonstrated a high sensitivity to water, and consequently to humidity. The addition of trace amount (0.01 wt%) of distilled water was highly sufficient to liquefy the organogel. Similarly, the sample organogel will often melt upon exposure to atmospheric humidity for a few hours. This behavior strongly proves that H-bonding is the driving force for the self-assembly and formation of the organogel. The action of water, and consequently humidity, in disrupting the organogel can then be ascribed to the disruption of the H-bonds by water molecules (see page 7, line 269-275).

  1. Figure 3 if possible gives data all the points of nm (350-650), then the fig will appear very smooth.

Response: Thanks for the reviewer recommendation. The wavelength (nm) already given in the range between 350-650 nm.

  1. If the author adds absorption maxima for each temperature in Fig. 3, it will increase the information from Fig. 3.

Response: The absorption maxima were added to each temperature in Figure 3.

  1. The authors gave a very nice mechanism in scheme 3, if possible authors can give the graphical abstract as well.

Response: Scheme 3 was introduced as a graphical abstract in a separate file.

  1. How many times/cycles the same solution shows reversibility for hot/cool conditions.

Response: As shown in Figure 5, there are 12 cycles reported the same solution showing reversibility for hot/cool conditions (see page 10, line 316).

  1. 5 legend is not clear at all, what is meant by different temperature values, need to mention it very clearly.

Response: The caption of Figure 5 was revised (see also page 10, line 313-321). Figure 5 describes the changes between the absorption maximum wavelengths of UTCFH (0.06 wt%) at 442 and 535 nm at the temperature values 44 and 63 °C, respectivley; pH value was adjusted at ~6.65. To inspect the reversibility of this UTCFH against temperature changes, the temperature of UTCFH in DSMO solution at pH = ~6.65 was changed back and forth by increasing the temperature from 25 to 65°C while recording the maximum absorption wavelength at 44 and 63 °C. The solution was then left to cool back to room temperature at 25°C to regenerate the organogel. This procedure was repeated twelve times while recording the absorption spectra for each cycle. The stability of the maximum absorption wavelengths at 442 and 535 nm, was recorded and the results are depicted in Figure 5. No significant changes in the maximum absorption wavelengths were monitored; at 442 and 535 nm. Thus, it is apparent that UTCFH exhibited high stability and high reversibility without fatigue.

  1. Have authors identified the mechanism with some structural validation, means structural changes in orange-red in urea, deprotonated hydrazone anion, and extended conjugation (purple) adducts

Response: the mechanism was proved by UV-Vis spectra (see page 8, line 279-290) as well as IR and 1H NMR spectra (see page 11, line 328-339; see also supporting information). The deprotonated nitro-substituted hydrazone anion form was proved by both of IR and 1H NMR spectra. The 1H NMR spectra was recorded at both room temperature (25°C) and 65°C to confirm that the secondary NH hydrazone peak (at 12.13 ppm) disappeared upon increasing temperature to 65°C (Figures S1-S5 and Figures S7-S8). Additionally, the signals at 8.58 (=C-H), 8.38 (aromatic CH), 8.23 (aromatic CH), 7.65 (aromatic CH) and 1.82 (aliphatic CH) demonstrated slight shifts to 8.50, 8.31, 8.20, 7.66 and 1.84, respectivley. The distinctive secondary NH hydrazone peak (at 3256 cm-1) was found to disappear in the IR spectrum of the deprotonated hydrazone anion form (Figures S6 and S9). To study the IR spectrum of the hydrazone anion form, the solid nitro-substituted hydrazone chromophore was dissolved in a mixture of triethylamine/acetone (1:1) and then air-dried under ambient conditions. The peaks at 2226 (CN), 1577 (C=N), 1509 and 1324 (NO2) showed some slight shifts to 2210, 1587, 1509 and 1328, respectivley.

  1. This is a nice piece of work, well written, well presented, and highly important study.

Response: Thanks for the reviewer feedback.

Round 2

Reviewer 1 Report

The authors have successfully addressed all my comments. The manuscript is interesting and worthy of publication. I m a lit bit confused by response «The temperature of the mixture was then raised from room temperature (25 °C) to 65 °C at a heating rate of 10 °C/min” instead of text in the revised version, where mentioned another speed  “(25 °C) to 65 °C at a heating rate of 2 °C/min”. Please, synchronize all parts of manuscript.

Author Response

Point-by-point response to Reviewers’ comments

Comments and Suggestions for Authors

The authors have successfully addressed all my comments. The manuscript is interesting and worthy of publication. I am a lit bit confused by response «The temperature of the mixture was then raised from room temperature (25 °C) to 65 °C at a heating rate of 10 °C/min” instead of text in the revised version, where mentioned another speed  “(25 °C) to 65 °C at a heating rate of 2 °C/min”. Please, synchronize all parts of manuscript.

Authors’ response: Thanks for the reviewer comments. We apologize for the typographical error in the response file. The correct heating rate should be “(25 °C) to 65 °C at a heating rate of 2 °C/min” as previously described in manuscript.

Reviewer 3 Report

In this revision the author have addressed some of the concerns of this reviewer, however authors still have not provided the gas adsorption isotherm data and the model fit curves to the adsorption isotherm data. 

Morever, since this reviewer was given more time to analyze the manuscript more close this time around, it is also clear that there is a substantial amount of water in the 1H NMR sample that the authors are using to prove the temperature driven deprotonation of the amine. With water present in the NMR sample, the amine proton will be in fast exchange with the water protons. This often results in the disappearance or broadening of amine protons in NMR spectra in the presence of water. This is clearly the case in the NMR spectrum taken at ambient conditions, the amine signal is barely visible, At higher temperatures, for example, at 65 oC, the amine proton exchange with water occurs at a much faster rate. This will most certainly result in the disappearance of the amine proton peak as observed in the NMR spectra measured at higher temperature by the authors. For this experiment to verify that temperature induced amine deprotonation is occurring by proton NMR spectroscopy, the experiment must be carried out in aprotic solvents and in the absence of water or other protic solvents. That is, the sample and NMR solvent should be anhydrous!

This reviewer must AGAIN suggest for this manuscript to be rejected as the authors have blantantly disregarded the call for the presentation of experimental data with regards to the adsorption isotherm experiments that the authors claim was conducted. This reviewer has not called into question the validity of the claims asserted by the authors based on the 1H NMR data.  

Author Response

Manuscript ID: chemosensors-964161

Title: Facile development of novel reversible colorimetric thermometer using urea organogel embedded with thermochromic hydrazone chromophore

Authors: Tawfik Khattab, Mehrez E. El Nagar, Meram Abdelrahman, Ali Aldalbahi, Mohammad Rafe Hatshan

Comments and Suggestions for Authors

In this revision the author have addressed some of the concerns of this reviewer, however authors still have not provided the gas adsorption isotherm data and the model fit curves to the adsorption isotherm data. 

Response: Again and as we mentioned in the manuscript, the surface area was determined using Quantachrome TouchWin software version 1.21 (new version). Brunauer-Emmett-Teller (BET) technique was employed to determine the specific surface area. This is a new version of software and the value of surface area was automatically calculated by the instrument. Thus, there is no equation needed or introduced by the instrument. Bellow is a reference published in high rank journal studying the surface area under the same methodology and instrumentation without provided the gas adsorption isotherm data and the model fit curves to the adsorption isotherm data.

  • Krejčová, Ludmila, Terza Leonhardt, Filip Novotný, Vilém Bartůněk, Vlastimil Mazánek, David Sedmidubský, Zdeněk Sofer, and Martin Pumera. "A Metal‐Doped Fungi‐Based Biomaterial for Advanced Electrocatalysis." Chemistry–A European Journal 25, no. 15 (2019): 3828-3834.

Also, I would like to get into the reviewer attention that the main target of the current research manuscript, as demonstrated in both manuscript title and abstract, is the development of novel reversible colorimetric thermometer using urea organogel embedded with thermochromic hydrazone chromophore. Thus, the psuedo first order or pseudo second order kinetics, as well as the gas adsorption isotherms and model fit data to the adsorption isotherms to evaluate whether it is a chemi-adsorption or physi-adsorption; monolayer adsorption or multilayer adsorption can be studied in a future work. 

Morever, since this reviewer was given more time to analyze the manuscript more close this time around, it is also clear that there is a substantial amount of water in the 1H NMR sample that the authors are using to prove the temperature driven deprotonation of the amine. With water present in the NMR sample, the amine proton will be in fast exchange with the water protons. This often results in the disappearance or broadening of amine protons in NMR spectra in the presence of water. This is clearly the case in the NMR spectrum taken at ambient conditions, the amine signal is barely visible, At higher temperatures, for example, at 65 oC, the amine proton exchange with water occurs at a much faster rate. This will most certainly result in the disappearance of the amine proton peak as observed in the NMR spectra measured at higher temperature by the authors. For this experiment to verify that temperature induced amine deprotonation is occurring by proton NMR spectroscopy, the experiment must be carried out in aprotic solvents and in the absence of water or other protic solvents. That is, the sample and NMR solvent should be anhydrous!

Response: Thanks for the reviewer comment. However, we would like to get into the reviewer attention that the tricyanofuran hydrazone chromophore is completely insoluble in water, which was already demonstrated in the experimental section that we rinsed the tricyanofuran hydrazone chromophore with distilled water. Thus, it is impossible for the NH exchange to occur.

This reviewer must AGAIN suggest for this manuscript to be rejected as the authors have blantantly disregarded the call for the presentation of experimental data with regards to the adsorption isotherm experiments that the authors claim was conducted. This reviewer has not called into question the validity of the claims asserted by the authors based on the 1H NMR data.  

Response: The authors believe that this reviewer is seeking to attack the credibility of the authors using unscientific claims just to find a way to reject this manuscript. As an evidence, this reviewer claims in every revision round new unscientific claims that he did not request before. The reviewer said aggressively in a previous revision round that our data is fabricated which is absolutely unethical accusation, which reflects his intention to attack the credibility of the authors seeking to find a way to reject this manuscript. I believe that a reviewer should normally and ethically ask for a certain claim and then wait for the author’s rebuttal response.